# A Review on Fast Tomographic Imaging Techniques and Their Potential Application in Industrial Process Control

**DOI:** 10.3390/s22062309

**Published:** 2022-03-16

**Authors:** Uwe Hampel, Laurent Babout, Robert Banasiak, Eckhard Schleicher, Manuchehr Soleimani, Thomas Wondrak, Marko Vauhkonen, Timo Lähivaara, Chao Tan, Brian Hoyle, Alexander Penn

**Affiliations:** 1Institute of Fluid Dynamics, Helmholtz-Zentrum Dresden-Rossendorf, Bautzner Landstraße 400, 01328 Dresden, Germany; e.schleicher@hzdr.de (E.S.); t.wondrak@hzdr.de (T.W.); 2Institute of Power Engineering, Technische Universität Dresden, 01062 Dresden, Germany; 3Institute of Applied Computer Science, Lodz University of Technology, Stefanowski 18, 90-937 Lodz, Poland; laurent.babout@p.lodz.pl (L.B.); robert.banasiak@p.lodz.pl (R.B.); 4Engineering Tomography Lab (ETL), Electronic and Electrical Engineering, University of Bath, Bath BA2 7AY, UK; m.soleimani@bath.ac.uk; 5Department of Applied Physics, University of Eastern Finland, P.O. Box 1627, 70211 Kuopio, Finland; marko.vauhkonen@uef.fi (M.V.); timo.lahivaara@uef.fi (T.L.); 6Tianjin Key Laboratory of Process Measurement and Control, School of Electrical and Information Engineering, Tianjin University, Tianjin 300072, China; tanchao@tju.edu.cn; 7School of Chemical and Process Engineering, University of Leeds, Leeds LS2 9JT, UK; b.s.hoyle@leeds.ac.uk; 8Institute of Process Imaging, Hamburg University of Technology, Denickestraße 17, 21073 Hamburg, Germany; alexander.penn@tuhh.de

**Keywords:** process tomography, tomographic sensors, image reconstruction, industrial process monitoring and control

## Abstract

With the ongoing digitalization of industry, imaging sensors are becoming increasingly important for industrial process control. In addition to direct imaging techniques such as those provided by video or infrared cameras, tomographic sensors are of interest in the process industry where harsh process conditions and opaque fluids require non-intrusive and non-optical sensing techniques. Because most tomographic sensors rely on complex and often time-multiplexed excitation and measurement schemes and require computationally intensive image reconstruction, their application in the control of highly dynamic processes is often hindered. This article provides an overview of the current state of the art in fast process tomography and its potential for use in industry.

## 1. Introduction

Control and automation systems are an integral part of today’s industrial production. At the lowest level, a control system may consist of sensors which obtain status data from the process, an analogue or digital control circuit which derives a control action for an actuator, and the actuator itself, which changes the process state by manipulating one or more process variables. The aim of control is to keep the process output within a permissible range around a predefined value. Examples of simple control systems in the process industry include temperature control of ovens, where the controlled variable is temperature, or level controllers in vessels, where the level is controlled by, for example, a hydrostatic pressure sensor and valve actuation. Such sensors may be described as satisfying a one-dimensional control requirement where a single point of measurement is selected to reasonably represent the process state. More sophisticated control systems may involve multiple sensors or actuators and complex, e.g., hierarchical, controller structures.

In recent decades, sensor technology has undergone tremendous development in many respects. The degree of miniaturization and integration has increased significantly, and modern sensors often measure multiple variables, can be self-calibrating, and operate in sensor networks. Imaging sensors are another direction. For example, video and infrared cameras are now very inexpensive and can be used as sensors in production. Their great advantage is that they are very versatile, since their function is determined by the image processing software used. Other types of imaging sensors are tomographic sensors. Unlike camera sensors, they reconstruct images from multiple integral measurement data, often obtained from transmission, emission, or boundary measurements on an object. Computed tomography has been known since the 1970s and was widely used first in medical diagnostics [1], and later in non-destructive testing [2]. Since the 1990s, the so-called process tomography has found its way into industrial process diagnostics [3,4]. It is often referred to as industrial process tomography (IPT), although is also widely used for flow measurement in, e.g., research labs. The ability of tomographic systems to image the inside of a body without penetrating the body has become a game changer in the control of critical processes in basic industries. Most industrial processes in the production of, for example, chemicals, metals, raw materials, food, or pharmaceuticals involve complex flows of materials from unit to unit, such as from a reactor to a distillation column or from a crusher to a silo or from a blast furnace to a mold. Process conditions can be very harsh in terms of pressure, temperature, corrosion, abrasion, etc., and sensor maintenance is often impossible or expensive. Therefore, non-intrusive sensors such as tomographic sensors are considered the best candidates.

With respect to process diagnostics, a number of tomographic imaging techniques have been developed over the past three decades, and there is a wealth of literature on various tomography principles and applications [3,4]. Thereby, process tomography can be divided into two lines. So-called hard-field tomography techniques have emerged from their medical counterparts, e.g., X-ray tomography, positron emission tomography, and magnetic resonance imaging. The other line is so-called soft-field tomography. Most prominent are electrical resistance (ERT) and capacitance (ECT) tomography, which reconstruct the conductivity or permittivity distribution of an object from electrical boundary measurements, and wire-mesh tomography (WMS), which obtains conductivity or permittivity distributions in gas-liquid flows in a direct way. Magnetic inductive tomography techniques have been devised for electrolyte and liquid metal flows. They reconstruct resistivity distributions from ac current excitations via boundary coils. Most recently, contactless inductive flow tomography (CIFT) has been invented, which obtains 3D velocity fields from remotely measured disturbances of an applied magnetic field pattern. Microwave (MWT) and ultrasound tomography (UST) measure wave attenuation along multiple transmission paths through an object and yield acoustic impedance or dielectric loss distributions. A reason for the existence of so many different modalities is their different capabilities to obtain certain process parameters. With respect to technical maturity it can be stated that ERT, ECT, and WMS have achieved TRL 7 to TRL 9, while more recently developed MIT, CIFT, MWT, and UST have TRL 6 or lower.

Process control typically requires control loops with response times of a few seconds, but often less. Here, tomography has an obvious disadvantage in that it requires multiple data to produce a single image. These data are often generated by time-serial excitation schemes, which have a negative impact on the acquisition time per image. In addition, tomography relies on a rather expensive computational solution to an inverse problem. Therefore, the use of tomographic sensors in industrial control is not very mature, and the published concepts often refer to very specific problems and solutions or are rather conceptual approaches with perhaps some laboratory demonstrations [5]. Meanwhile, with the availability of fast processors and highly parallel digital computer structures, the latter is no more a general constraint.

Tomographic sensing inherently provides multi-dimensional data for processes in which one-dimensional sensors, such as temperature and pressure exemplified above, are unable to represent a process that involves multiple components and/or distributed properties where one-dimensional data are inadequate. A dynamic image representation is likely to be of interest to process designers using a pilot-scale trial. However, for full-scale processes, image-based data must be transformed, or interpreted, into a useful process variable transmitted to an overall plant controller. For example, an estimation of the state on a projected batch process trajectory. The resulting composite process, including tomographic sensing, image generation, and interpretation, is increasingly commonly combined, for example, using neural approaches, to provide fast direct estimation from tomographic sensor to interpreted control data [4].

This article has the mission to bring the state of the art of fast process tomography to the attention of the reader and discuss recent achievements in terms of their usability for fast industrial process control. This endeavor comes with the difficulty that there exist a vast multitude of different tomographic imaging modalities and it is hardly possible to discuss their physical principles as well as electronic and computational particularities here in full detail. For that, the reader may be referred to some fundamental articles, which are given in the respective sections of the textbooks referred to above.

## 2. Electrical Resistance and Electrical Capacitance Tomography

Electrical resistance and capacitance tomography modalities (ERT and ECT, respectively) are among the most popular process tomography techniques. These imaging methods are sensitive to passive electrical properties, such as electrical conductivity for ERT and permittivity for ECT, making them potentially suitable for many industrial applications such as chemical engineering, food industry, or energy engineering. Both techniques use electromagnetic fields to sense electrical properties of the object under investigation. This usually necessitates the following components: (1) a set of electrodes on the boundary of the vessel, (2) a data acquisition unit that controls how a generated signal is injected sequentially into excitation electrodes and collected by others in a specific measurement scheme, and (3) a computing unit responsible for data processing of the collected signals [6]. Figure 1 shows typical schematics of ERT and ECT systems. The figure also illustrates industrial applications where the input of ECT/ERT is of added value: monitoring of density distribution and velocity estimation to characterize flow regimes [7] and phase separation of two-phase flow [8]. Another industrial application where ERT and ECT showed promising results is the measurement of moisture distribution in a material [9,10]. Both modalities can also be coupled to investigate processes where three phases are mixed, such as in the case of gas-oil-water flow [11,12].

Companies and academic institutions propose off-the-shelf ECT and ERT tomography equipment with a temporal resolution that spans the 10 Hz–5 kHz range depending on the designed electronics and data acquisition schemes [7,13,14,15]. While ECT relies on voltage injection, ERT systems are designed with electronic hardware that supports either current injection and voltage measurement (CV mode) or the other way around (VC mode). The latter is, in practice, simpler to design, allows a multiple excitation scheme, and can achieve the same level of accuracy as CV-based systems, provided that good contact between electrodes and the conductive medium is guaranteed [16]. The VC systems can adapt to a large conductivity range, from demineralized water to strong ionic solutions, thanks to output conditioning circuits [8,13]. Inspired by ECT sensors, alternative ERT hardware designs also considered avoiding sensors directly in contact with conductive but potentially corrosive media, such as the so-called capacitively coupled electrical resistance tomography (CCERT) system primarily designed to measure the conductivity distribution of two-phase flow [17]. Different strategies were also employed to increase the number of electrode pair measurements to improve image resolution in the 2D or the 3D space, especially for ECT. This includes the development of rotatable or 3D sensors [18,19,20]. Comprehensive reviews for ECT sensors in circulating fluidized beds can be found in [21,22], and reviews for ERT sensors in chemical engineering can be found in [23].

Electrical tomography systems provide cross-sectional maps of the respective electrical property by use of image reconstruction algorithms. Cui et al. reviewed the most important direct and indirect image reconstruction strategies developed in the last three decades [24], with some of them being implemented in commercial software [25]. Considerable effort has been made to obtain the reconstructed images with the highest possible accuracy. However, ERT and ECT images, like other “soft-field” tomography images, are impacted by the non-linear characteristic of the electromagnetic field and the ill-posedness of the measurement scheme. As high reconstruction accuracy also causes some computational load, ERT and ECT are often used as off-line diagnostic tools. Nonetheless, the popular trend of artificial intelligence and machine learning recently opened new avenues to deal with non-linearity and ill-posedness issues [26,27,28,29]. These approaches necessitate a knowledge base of the process under investigation, which relies on either experimental or simulated trained datasets. Once the training process is completed, reconstruction and information extraction can be performed very fast, allowing real-time imaging and potential process control, respectively.

## 3. Multi-Spectral and Spectro-Tomography

IPT instruments use an excitation mode(s) with a specified energy level to interact with process materials to produce sensed response data used to estimate content information. This often complex interaction is summarized by the term *contrast*. For example, a first-generation (1G) electrical capacitance unit, featuring single mode/energy level (1G-SM-SE), is adequate where a high-contrast material is imaged within a low-contrast material; for example, as pioneered in 1990 for oil in gas by Dickin, Hoyle et al. [30]. Many current IPT systems successfully exploit 1G operation. Where two materials are jointly of interest, multi-mode (2G-MM) instruments (such as acoustic and electrical) have been developed, but are complex in sensing field and engineering [31]. Figure 2a exemplifies electrical energy level (frequency) contrast characteristics (at stated temperatures) typical of many pharmaceutical compounds. Many industrial processes feature reagents chemically transformed into a product, ideally with sensing data required for all components, where simple 1G IPT systems are likely to fail. Figure 2b shows simplified contrast characteristics of reagents A and B and product C, with excitation and responses for fixed level (frequency) excitation. The top row of Figure 2c shows the process state at three progressive sample times. *Multi-spectral* excitation may be applied sequentially, reducing temporal resolution, and exacerbated by the need for a settling delay for each injection transient. Use of additive signals avoids this issue but requires response filtering to extract three material datasets. Typical IPT products, such as the ITS P2+, support this operation [32]. The three response datasets are used to generate the tomographs illustrated in the center rows of Figure 2c for frequencies 20, 50, and 80 kHz, falsely colored in red, green, and blue scales. Clearly, each 1G-SM-SE (20, 50, 80 Hz) individually fails to generate reliable self-standing data, due to the joint multiple component characteristic. Crude fusion for each corresponding pixel (or local group) across the three tomographs reveals the most likely material identification for the corresponding point, as illustrated in the generated interpretation of Figure 2d. This method is probably used occasionally in more complex processes.

A more powerful and elegant method proposed by Hoyle and explored with Nahvi [33] has been investigated to synthesize a formal *Multi-spectral* method in which tomographic and corresponding spectroscopic material identification data are jointly generated. Wideband linear compressed frequency *chirp* excitation is deployed, where frequency sweeps through the range over a short time after a single initial transient. The resulting response projection corresponds to a single time interval, but contains coherent data for all frequencies in the selected band. The overall dataset can typically be collected in a period comparable with a 1G-SE IPT. A tomograph can then be computed (or *extracted*) for any frequency, based upon the single interval projection data. Multi-point *spectral fingerprints*, including mixture concentrations, may then be deployed in interpretation to jointly estimate temporal tomographic process data with identified spectroscopic characteristics [34]. Figure 2d illustrates the overall method in use. Exploitation of this 2G SM ME method will require powerful integrated digital processing but is likely to be used in future applications where detailed knowledge of complex processes is required for Industry 4.0 control needs.

## 4. Wire-Mesh Sensors

A wire-mesh sensor can be considered a tomographic sensor because it generates cross-sectional images of a flow. However, it does not involve image reconstruction. Instead, the images are obtained directly by measuring an electrical property of the fluid in a plurality of volume elements defined by the electric field in the crossing points of a wire electrode grid. Figure 3 illustrates the principle for a sensor mounted in a pipe. Equidistantly arranged wires are stretched across the cross-section of the pipe in two planes. In one plane are the transmitter wires, in the other plane the receiver wires. The distance between the planes is about 2 mm. The measurement protocol is to excite each transmitter wire in turn with a voltage signal and measure the electric current on all receiver wires in parallel, while holding the unexcited transmitter wires at ground potential. After one turn of the excitation, a complete map of the impedance values for the crossing points is obtained. The wire-mesh sensor was first introduced by Prasser et al. using a DC excitation and measurement scheme [35]. The images show the conductivity distribution in a flow. This makes the sensor suitable for the investigation of gas-liquid and liquid-liquid multiphase flows with at least one conductive liquid. Furthermore, the sensor can be used to obtain local flow velocities or to investigate global mixing in pipes and vessels, e.g., by using salt tracers. Da Silva et al. extended the principle towards capacitance [36] and impedance measurements [37] using an AC excitation and measurement scheme. Furthermore, the principle has been extended to temperature measurement [38] and velocity measurement employing a thermal anemometry approach [39].

Since impedance measurement in liquids can be very fast by its very nature, the wire-mesh sensor is a very fast tomographic imaging device. Commercial sensors come with a typical scanning speed of 10,000 frames per second but higher rates would be possible. Using such sensors for fast process control is hence possible if fast data processing is implemented. Kipping et al. [40] developed a fast fuzzy-logic-based data processing scheme implemented on FPGA hardware. Gas holdup and flow regime are derived from the windowed frequency distribution of the gas holdup in the cross-section. The method was further refined towards pseudo-dynamic operation by Wiedemann et al. using an extended fuzzy C-means clustering approach [41]. More recently and also reported in this special issue, wire-mesh sensors have been used to control inline fluid separation by sensing the flow conditions upstream of the splitter [42].

## 5. Magnetic Induction Tomography

Magnetic induction tomography (MIT) can produce images of the passive electromagnetic properties (PEP) of an object. Its function can be explained by the mutual inductance and electromagnetic eddy current models. MIT can be made with an array of coils acting as excitation and receiving coils. When injecting an alternating current into the excitation coil, a primary magnetic field is generated, which induces voltages in the measuring coils. These voltages in turn depend on the PEP distribution. The first use of MIT is known from geophysical applications [43], but in the last three decades it has been established for industrial process applications too [44]. Early industrial applications of MIT focused on its use in liquid metal diagnostics [45] and some theoretical studies focused on its possible use in metal solidification, both in the field of steel continuous casting. Application for metal flow imaging was further investigated in [46] and use in a real continuous casting solidification was demonstrated in [47]. The injection of argon gas and stoppers opening in submerged entry nozzles in steel production can be controlled via the liquid metal flow regime identified by the MIT sensors. MIT-based measurement of thickness of the solid/mushy/liquid phase in the hot casting stage in steel solidification can be used for control of casting speed as well as control of the flow of cooling water in the spray cooling stage. MIT for metal-based applications is dealing with materials with high electrical conductivity and operating at a low frequency below a few hundred kHz. Inspired by the medical applications of MIT [48], the low-conductivity and high-frequency MIT (generally around 10 MHz) was investigated for inline water cut measurement in the petrochemical sector. Much progress has been made in developing a better understanding of the MIT image reconstruction problem and in MIT hardware development. In [49], a multi-frequency MIT device was introduced, enabling better than 10 frames per second data collection. The image reconstruction in MIT is a non-linear ill-posed inverse problem [50], but for many real-time industrial applications, a regularized linear algorithm is more suitable. In [51], a spectral MIT system and algorithm show the extended capability of the MIT for metal type classification as well as a novel application of contactless temperature measurement. Due to its contactless nature and the possibility of use in high-temperature environments, MIT is a unique solution for various industrial applications; growth is expected on such an application. For imaging deep inside metal samples, efficient use of multi-frequency MIT both in terms of measurement speed and computational modeling can pose further scientific challenges which need to be resolved before new applications can be realized. Figure 4 shows an example of a process involving an MIT system for metal sample imaging. After solving the forward eddy current problem, the resulting model data are then compared with the experimental data, allowing verification of both models used and a verification of the MIT data acquisition system. The inverse problem can be solved by use of the traditional linear/non-linear inversion algorithm or with the aid of neural network approaches. MIT has a unique place in liquid metal applications due to its adaptability in harsh environments and also in multiphase flow imaging, especially due to its contactless nature.

## 6. Contactless Inductive Flow Tomography

The three-dimensional flow structure in electrically conducting melts can be reconstructed by the contactless inductive flow tomography (CIFT) [52,53,54,55]. The application area of CIFT includes all stages of liquid metal processing in metallurgy such as mixing, alloying, casting, and solidification. A prominent application is continuous casting of steel, which is used for 96% of the world’s steel production [56]. The complex flow regime in the mold of a continuous caster plays an important role in the quality of the produced steel regarding surface defects and inclusions [57]. The opaqueness of liquid steel prevents the use of established optical flow measurement techniques. Furthermore, the high temperature of 1500 °C of liquid steel requires measurement techniques without direct contact with the melt. CIFT relies on the measurement of the perturbation (secondary field) of an applied magnetic field (primary field) by the flow of the melt. The magnetic field sensors are located outside the fluid volume at a given distance. From this measurement, the flow structure is reconstructed by solving a linear inverse problem with an appropriate regularization technique to circumvent the non-uniqueness of the problem [58]. For a three-dimensional reconstruction, two applied magnetic fields in different directions are needed, while for a two-dimensional reconstruction, one applied magnetic field is sufficient. The reliable measurement of the magnetic field is challenging because the secondary magnetic field is 2 to 5 orders of magnitude lower than the primary magnetic field. Therefore, the magnetic field sensors as well as the AD converters have to cope with a dynamic range of 6 orders of magnitude and must provide a very linear response [59]. Usually, the strength of the applied magnetic field is about 1 mT and the flow-induced perturbation is in the order of 10–300 nT. The strength of the applied magnetic field cannot be arbitrarily increased because an alteration of the flow by Lorentz forces has to be avoided.

If the applied magnetic field is a static field, mainly Fluxgate probes are used [60,61]. These measurement tools are very sensitive to environmental noise and cannot be applied in an industrial environment. A more robust measurement can be achieved by employing an alternating applied magnetic field with a rather low frequency in the order of 1 Hz, such that the skin effect can be neglected. This allows for suppressing undesired signals from the environment [54,59,62]. For the precise measurement of the magnetic field, induction coils with 360,000 turns and a diameter of 28 mm are used [63]. The rejection of noise can be further increased by measuring the gradient instead of the magnetic field itself [64]. With this technique, CIFT is now applicable even in the presence of a static magnetic field in the order of 300 mT.

The development of CIFT started with the experimental validation of the technique using a cylindrical vessel, in which the flow of the eutectic alloy GaInSn was driven by a propeller [53]. Forty-eight Hall probes distributed around the vessel measured the secondary magnetic field for two applied magnetic fields in the vertical and horizontal direction. Typical velocities were in the order of 1 m/s and the ratio of the applied and the flow-induced magnetic field was about two orders of magnitude. Based on these results, CIFT was then adapted to a model of a continuous caster operated with the eutectic alloy GaInSn at room temperature [60]. Only one magnetic field in vertical direction was applied and the sensors were placed along the narrow faces of the mold. Figure 5a shows a sketch of the CIFT sensor arrangement at the mold with a cross-section of 140 mm × 35 mm and the reconstructed velocity field. The dynamics of the flow in the mold could be reconstructed for gas injection in the SEN and were in good agreement with the accompanying ultrasound Doppler velocimetry measurements [61]. By using an alternating applied magnetic field, the flow in the mold can be visualized even in the presence of a static magnetic field of an electromagnetic brake (EMBr) [62,63,64]. The sensor arrangement is shown in Figure 5b. While in these measurements the strength of the EMBr was not changed, the measurement system was recently extended to cope also with changes of the strength of the EMBr during the CIFT measurements [65]. The key aspect is the compensation of the effects of the ferromagnetic parts on the magnetic field measurements. In combination with a real-time reconstruction algorithm [65], CIFT is now able to act as a monitoring tool and as a sensor for a control loop. The scalability of CIFT was demonstrated by successful reconstructions of the flow structure in a larger mold with a cross-section of 500 mm × 100 mm of the LIMMCAST facility, which is operated with SnBi at 250 °C [66].

## 7. Microwave Tomography

In microwave tomography (MWT), a set of transmitting antennas is used to send electromagnetic waves in the microwave regime (~300 MHz–300 GHz) into the object and the scattered waves are collected with the same or a different set of antennas. Image reconstruction in MWT is based on the relationship between the transmitted microwaves and the medium of propagation, which is fully determined by the relative permittivity (dielectric properties) of the medium [67,68,69]. Over the years, the MWT has been applied in various fields, including medical imaging [67,68], non-destructive testing [69], geophysical prospecting [70], and security [71]. In industry, MWT has been widely utilized [72,73]; for example, in multiphase flow imaging [74] and detecting moisture distribution in microwave drying of porous materials [75,76,77].

A standard measurement protocol in MWT is to excite each transmitting antenna in turn with a microwave signal and measure the amplitude and phase on all receiver antennas. For obtaining the scattered-field data, two different experiments are needed; see Figure 6. First, the incident-field data *E*_inc_ is collected in the absence of the object and second, the total-field data *E*_tot_ is collected in the presence of the object. The scattered-field dataset *E*_sct_ is obtained by subtracting the incident-field dataset from the total-field dataset. The scattered-field data are used to reconstruct the complex permittivity distribution of the object. In practice, the antennas are connected, via an electronic switch (switching matrix), to a commercial/custom vector network analyzer (VNA) for acquiring multi-static and multi-view electric field measurements at single or multiple frequencies. A few examples of the MWT experimental setups are described in [78,79,80]. After the scattering data are collected and calibrated, the permittivity distribution of the object is estimated by an appropriate inversion algorithm. The algorithm needs to handle both non-linearity and ill-posedness of the problem. The non-linearity is commonly handled using iterative algorithms such as gradient-based methods [81,82,83,84], and the ill-posedness is usually accounted for by a Bayesian inversion approach [85,86] or by incorporating appropriate regularization techniques. Recently, neural-network-based approaches have also been developed [75,76,87]. These techniques enable fast image reconstructions even though the problem is non-linear.

The use of MWT in controlling industrial processes is still in its infancy. This might be because most of the current measurement devices and reconstruction algorithms, especially when multi-frequency data and non-linear reconstruction algorithms are used, do not reach the sub-second speed requirements of the control systems. Therefore, to meet the speed requirement of the control, strategies such as limited-band or single-frequency reconstructions, utilizing, for example, machine learning or linearized inversion schemes, can be adopted, in combination with a custom-built data acquisition setup [88,89,90]. In addition, MWT has several important features, such as safety due to the low-level operating power, good contrast, and completely contactless operation, which make it a potential choice for control applications in various industrial processes.

## 8. Ultrasound Tomography

Ultrasonic process tomography (UPT) utilizes the multiple transmission features of ultrasound in a medium [91]. At high frequency, ultrasound shows good directional propagation that makes the ultrasound tomography principle very similar to radiation tomography. However, ultrasound waves show complex interactions on the interface between two media of different acoustic impedance, such as reflection, transmission, refraction, and Doppler effect, making it a typical multi-mode tomography [92]. UPT has the advantage of being non-hazardous and non-intrusive and it even employs clamp-on use, which makes it an ideal choice for industrial process monitoring. UPT can be typically grouped into ultrasonic transmission tomography (UTT) and ultrasonic reflection tomography (URT) as shown in Figure 7. The time-of-flight (TOF) and sound attenuation along different ray paths are collected to reconstruct the spatial distribution of acoustic properties. For the URT, the reflected ultrasound waves due to the acoustic impedance difference between gas and liquid are collected. Given the reflection ray path for each transducer-receiver pair, the amplitude and the TOF extracted from the reflected signal can be used to accurately locate the inter-phase boundary and reconstruct the shape of the inclusions.

The imaging quality relies very much on the number of independent projections and the signal-to-noise ratio. Therefore, key developments for UPT focus mostly on an improved system design [93]. More transducers provide more projection data but reduce the data sampling rate because ultrasound propagation in fluid is much slower than electromagnetic wave propagation. Therefore a typical number of transducers is 16 or 32 [94]. Meanwhile, fan-beam transducers can help increase the number of receiving transducers under each excitation [95]. As such, Murakawa et al. presented 8 wide-angle transducers URT for bubbly flow imaging with a frame rate up to 1000 fps [96]. Both the UTT and URT have their own advantages and drawbacks. Specifically, UTT has better performance on the inclusions close to the boundary, while URT achieves better reconstruction quality in the center area. To fully utilize their advantages, a dual-mode UPT that combines UTT and URT by using only one array of ultrasonic transducers has been developed, in which both the ultrasound attenuation and TOF can be collected by all transducers under the “one-to-all” strategy to reconstruct the fluid distribution at a frame rate up to 625 fps with 16 transducers, which shows its potential for industrial multiphase flow imaging [97].

Typically, UPT is used to image gas-liquid distributions, such as bubbly flow [92], because gas and liquid have very distinctive acoustic impedances. It can also image fluid density distribution by reconstructing the TOF with UTT. This was, e.g., demonstrated for a batch crystallization process by Koulountzios [98]. By further employing multi-frequency ultrasound, UPT can also be used to image three-phase flows, such as of oil, water, and gas, by utilizing the ultrasound attenuation spectrum [99]. Additionally, other parameters such as temperature and flow velocity can also be reconstructed [100], which makes UPT a promising candidate for complex process imaging.

## 9. X-ray and Gamma Ray Tomography

X-ray computed tomography can be considered the archetype of all tomographic imaging techniques. G. N. Hounsfield introduced X-ray tomography to medical diagnostics in 1972 [101]. Since then, it has undergone tremendous development. Today’s commercial medical CT scanners can scan the human body at a maximum of two frames per second, allowing imaging of the interior of the body at a resolution of half a millimeter. Spiral CT and later the introduction of digital flat-panel X-ray detectors enabled 3D tomography. Later, when micro-focus X-ray tubes entered the market, X-ray imaging and tomography were established in the field of non-destructive testing and evaluation. However, ionizing radiation-based process tomography has been limited mainly to laboratory-scale scientific studies rather than industrial use in the process industry. One reason for this is the radiation hazard associated with X-ray and nuclear radiation sources. Another reason is the difficulty of producing dynamic images. While the availability of high-power X-ray tubes and synchrotron X-ray sources has made dynamic radiography possible since the 1980s, dynamic radiation tomography with sufficient spatial resolution remained difficult for a long time. A first step in this direction was taken with the commercialization of ultrafast cardiac X-ray tomography, which uses the principle of electron beam scanning [102]. This development was driven by the need to image a patient’s beating heart. The devices achieved frame rates of up to 20 per second. With the advent of fast spiral X-ray CT and cardiac MRI, these scanners gradually disappeared from hospitals due to their high investment and maintenance costs.

The development of dynamic scanning principles for multiphase flows was driven by the oil and gas industry’s need to measure oil, water, and gas separately during oil exploration. To this end, a high-speed gamma-ray tomograph has been developed at the University of Bergen, Norway, (Figure 8a) [103]. It comprises five discrete Am-241 sources ($1.85\times 10^{10}\;\mathrm{Bq}$, $E_{\gamma }=59.54\;\mathrm{keV}$) together with five line detectors, each consisting of 17 CdZnTe room-temperature semiconductor detectors. The detectors and sources are collimated so that each detector is in the fan of only one gamma-ray source. This enables a continuous transmission scan at the expense of a limited number of projections. The inverse limited-data problem is solved as a system of algebraic equations with an iterative solver. The scanner achieves frame rates of up to 278 frames per second.

Using multiple switched X-ray sources instead of gamma ray sources allows for denser sampling in terms of number of projections. However, this comes at the cost of more expensive equipment. X-ray sources are themselves expensive, but even more importantly, commercially available X-ray tubes typically do not have a fast switching function. For the latter, tubes must be equipped with special Wehnelt electrodes or shutter grids between the cathode and anode. Morton et al. [104] developed such a system with 12 X-ray elements, each producing 13 selectable X-ray spots in a line. Together with a fixed X-ray detector, this results in a tomography system with 156 projections and a frame rate of 50 frames per second. Misawa et al. [105] built a tomography scanning system with 18 switchable miniature X-ray tubes (Figure 8b). The system achieved 263 frames/s. It was later expanded to 60 X-ray tubes and 2000 frames per second by Hori et al. [106]. Another fast X-ray tomography system with multiple sources is located at Delft University, The Netherlands. It has three X-ray sources and three 2D flat panel detectors of 1524 × 1548 pixels. At full resolution, the imaging speeds are 22 frames per second. However, when only a part of the detector area is used, up to 200 frames per second is possible. These detectors allow for 2D projections of the measured object. These can be combined to make a 3D reconstruction [107].

Ultrafast electron beam tomography enables simultaneous fast and high-resolution X-ray tomography. At Helmholtz-Zentrum Dresden-Rossendorf, Germany, such a system has been developed for scanning multiphase flows (Figure 8c). It consists of an electron beam generator with an electromagnetic deflection system that allows scanning of a 150 kV, 10 kW electron beam across a circular tungsten target in two planes. A fixed dual-ring detector with small room-temperature semiconductor pixels scans the X-ray flux at 2 MHz frequency. The frame rate is up to 8000 frames per second and the spatial resolution is 1 mm [108].

Despite the great progress and potential of fast radiation tomography, applications in process control are still very rare due to the difficulties mentioned above. A remarkable example was presented by Windisch et al. They developed a control strategy for the Rossendorf Ultrafast X-ray system to track structures in the flow, such as gas bubbles or particles, in real time by moving the scanner in the axial direction, e.g., along the axis of a tube [109]. Of course, it is not the process itself that is controlled, but the scanning system, in order to virtually extend the axial field of view.

**Figure 8 sensors-22-02309-f008:**
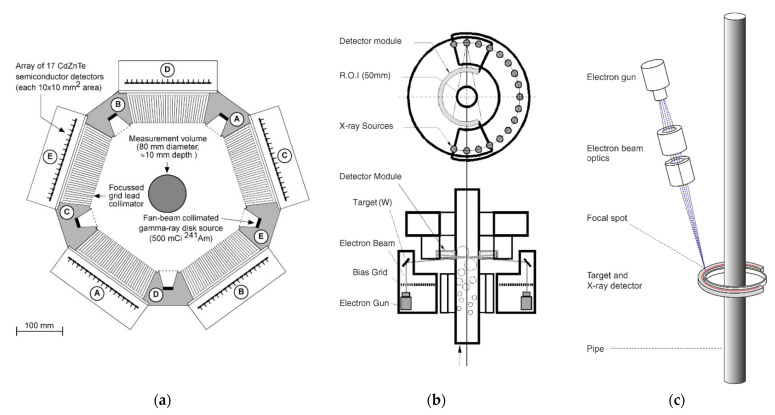
Fast radiation tomography systems: (**a**) University of Bergen fast gamma ray tomography scanner [103] (reproduced with permission from G.A. Johansen et al., *Appl. Radiat. Isot.*
**2010**, *68*, 518–524; published by Elsevier, 2010). (**b**) Fast X-ray tomograph with 18 switched X-ray tubes of the National Institute of Advanced Industrial Science and Technology Japan [105] (reproduced with permission from H.-M. Prasser et al., *Flow Meas. Instrument*
*16*, 73–83; published by Elsevier, 2005). (**c**) Ultrafast electron beam X-ray scanner ROFEX of the Helmholtz-Zentrum Dresden-Rossendorf [108] (reproduced with permission from F. Fischer et al., *Nucl. Eng. Des.*
**2010**, *240*, 2254–2259; published by Elsevier, 2010).

## 10. Magnetic Resonance Imaging

Magnetic resonance imaging (MRI) is a tomographic modality that allows quantifying a wide variety of sample properties in industrial process systems. These properties include the local density and distribution of phases, velocity, diffusion, temperature, and chemical composition and reactions of the analyzed samples. Moreover, MRI can detect system-specific properties such as fluidization or shock waves in gas-solid systems [110].

MRI is based on the physical phenomenon of nuclear magnetic resonance (NMR) discovered by Rabi et al. in 1938 [111]. In 1973, Lauterbur [112] combined NMR with spatial encoding gradients, hence laying the foundation for MRI. NMR spectroscopy is widely used in the chemical sciences to investigate the structure and interactions of atoms, molecules, and proteins while its imaging counterpart MRI has become an indispensable tomographic tool in clinical diagnostics. Both NMR and MRI are based on the principle that if nuclei of non-zero nuclear spin (most prominently ^1^H, but also ^19^F, ^13^C, and several others) are brought into an external magnetic field, the nuclear spins of the material do partially align with the field. Moreover, the spins precess at the Larmor frequency $\omega_{L\;}=\gamma \;B_{0}$, which depends mainly on the magnetic field strength $B_{0}$ and the gyromagnetic ratio *γ* of the nucleus. By radiating the sample with electromagnetic waves at the resonance frequency $\omega_{L}$, the nuclear spins within the sample are excited into a higher energy state. After the excitation, the spins relax into their equilibrium state while emitting detectable electromagnetic waves which carry information about the sample. In recent decades, MRI has attracted increasing interest from the engineering community because of the wide range of contrasts it can produce. A comprehensive introduction into the fundamentals of MRI is provided in the textbook by Haacke [113]; a brief introduction of MRI in the field of industrial tomography is provided by Sederman [114].

Figure 9a shows a clinical MRI scanner suitable to study a variety of engineering systems. Figure 9b displays different MRI measurement types, while Figure 9c indicates their respective current limits in temporal resolution and provides an estimate about the potential of further speeding up these measurement types using known acceleration techniques.

In MRI, data are not acquired in the image domain, but in reciprocal *k*-space, which is reconstructed to an MR image—in the simplest case, by applying a Fourier transform. In contrast to other tomographic modalities (e.g., X-ray computed tomography) MR data are sampled in a sequential manner, causing generally rather low temporal resolutions, a traditional shortcoming of MRI. During recent decades, however, numerous acceleration techniques have been developed, improving the temporal resolution of MRI by several orders of magnitude. These acceleration techniques include: (i) single-shot readouts such as echoplanar [118] and spiral imaging, (ii) parallel imaging involving multichannel signal receiver arrays [119], and (iii) compressed sensing [120]. Most of these acceleration methods can be utilized readily in engineering applications. Moreover, most engineering applications allow optimizing the MR signal lifetime of the sample by adding contrast agents.

In most previous uses of MRI in the engineering field, NMR spectrometers equipped with tomographic probe heads have been used. These systems limit the sample diameter to about 60 mm, which is relatively small for most industrial process systems. The use of full-body clinical MR platforms (Figure 9a) allows sample diameters of 200 mm and more. Recently, the combination of a custom-built MR detector array on a clinical MR platform, time-efficient single shot readouts, and engineered MRI-active particles, has improved the temporal resolution of granular velocity measurements in gas-solid systems by more than four orders of magnitude [110]. This methodology has been further applied to gas-liquid and liquid-solid systems and opens a wide range of possibilities for real-time investigation of multiphase flow systems in an engineering context.

A central limitation of MRI in the field of engineering is the fact that samples and container materials need to be non-ferromagnetic and ideally electrically non-conductive. Most commonly, glass, PEEK, or PMMA are used as container materials. MRI can be used to image liquids, gases, and solids. The highest spatial and temporal resolutions are achieved for liquids. Contrast agents such as paramagnetic salts can be dissolved into the sample to modify relaxation times and ultimately increase the signal-to-noise ratio (SNR). In gas MRI, the signal intensity is reduced by the number of molecules per volumetric unit in comparison with liquids. Therefore, researchers often work at increased gas pressures or employ hyperpolarization techniques to increase SNR, such as parahydrogen-induced hyperpolarization [121]. To study the dynamics of solids (e.g., in gas-solid flows), the solids particles are usually filled with liquids, which in turn are well suited for imaging.

Despite its unparalleled versatility and the recent improvements in temporal resolution, MRI is still only a niche application in the field of industrial process diagnostics. The high system cost and complexity are the largest barrier to more widespread use. Hence, available MRI systems should be shared more extensively among researchers, similar to electron microscopes or beamline facilities.

## 11. Conclusions and Outlook

Process tomography has long been used mainly to monitor industrial processes or to facilitate the analysis of multiphase flows in laboratory environments. While its inherent non-intrusive nature when capturing material flows in pipes and vessels has made it a promising technology for process control, very few practical examples exist. A major reason for this is the need for image reconstruction and parameter extraction, which requires computational time that may not be available for fast control problems. In some modalities, such as ultrasound tomography and MRI, temporal resolution is also limited due to the physics of signal creation or the time-multiplexed sensing schemes used. In the future, it is expected that the number and type of applications of tomographic sensors in industrial control will increase. This is favored by three technological developments. (1) Computer hardware is becoming more powerful in terms of speed and data throughput. In particular, GPU systems are now widely used to accelerate complex image reconstruction and analysis tasks. Other important hardware developments include gigahertz data transmission links and 5G technology. (2) Machine learning strategies are now mature and enable rapid image analysis and classification. Combined with ML-based image reconstruction, this results in a powerful approach for fast decision making in control loops. (3) There is an increasing number of general image sensors used in industrial diagnostics and control. Optical and infrared cameras do not require image reconstruction, but their application will continue to drive the development of fast hardware and software components for imaging-based industrial control systems.

In this paper, a brief overview of the current capabilities and development directions of the major process tomography classes has been presented. It can be stated that tomography hardware, i.e., transducers, electronics, process compatibility, standard image reconstruction, and data processing have by far reached TRL 7 and higher in terms of process diagnostics. The best candidates for industrial process control remain technologies with lower complexity and hazard, such as electrical, magnetic, ultrasound, and microwave tomography, although tradeoffs must be made in terms of spatial resolution. For process control, the future must focus on rapid parameter extraction with or without image reconstruction, as well as appropriate controller structures for multidimensional data. Table 1 summarizes in a very condensed way the main properties of the considered process tomography techniques with respect to their applicability in industrial process monitoring and control.

## Figures and Tables

**Figure 1 sensors-22-02309-f001:**
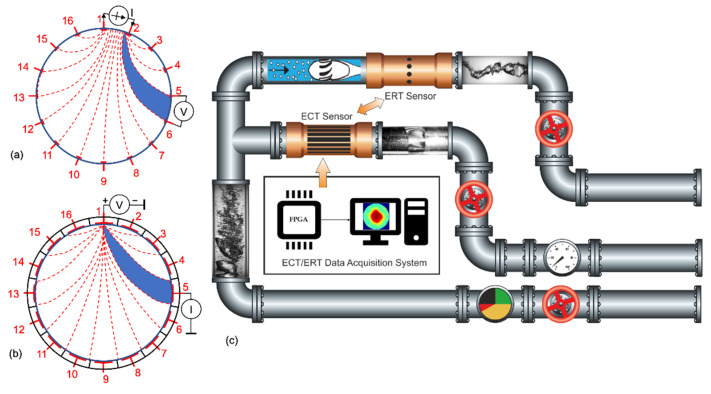
(**a**) Schematic of an ERT system. (**b**) Schematic of an ECT system. (**c**) Illustration of ECT and ERT systems mounted on flow installations. Red dashed lines in (**a**,**b**) delimit iso-potential areas represented by blue surfaces between the injection and measurement electrodes.

**Figure 2 sensors-22-02309-f002:**
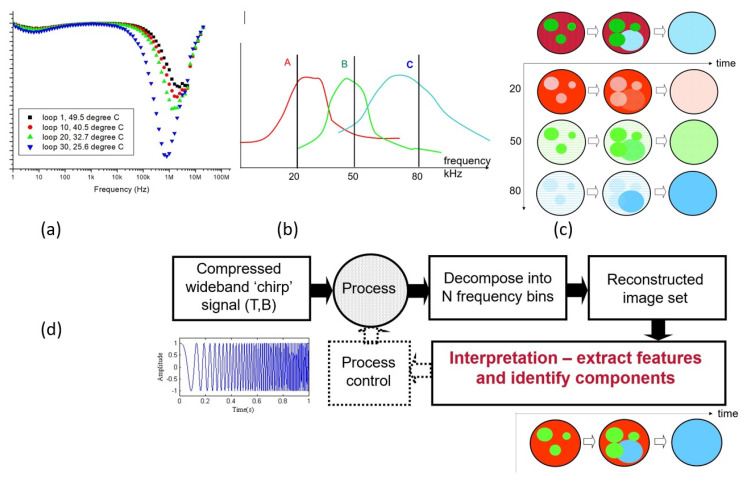
Principle of spectro-tomography (**a**) Example material spectral contrast characteristic. (**b**) simplified process contrast characteristic for reagents A, B and product C. (**c**) process state at 3 sample times. (**d**) composite excitation, reconstruction and interpretation processing.

**Figure 3 sensors-22-02309-f003:**
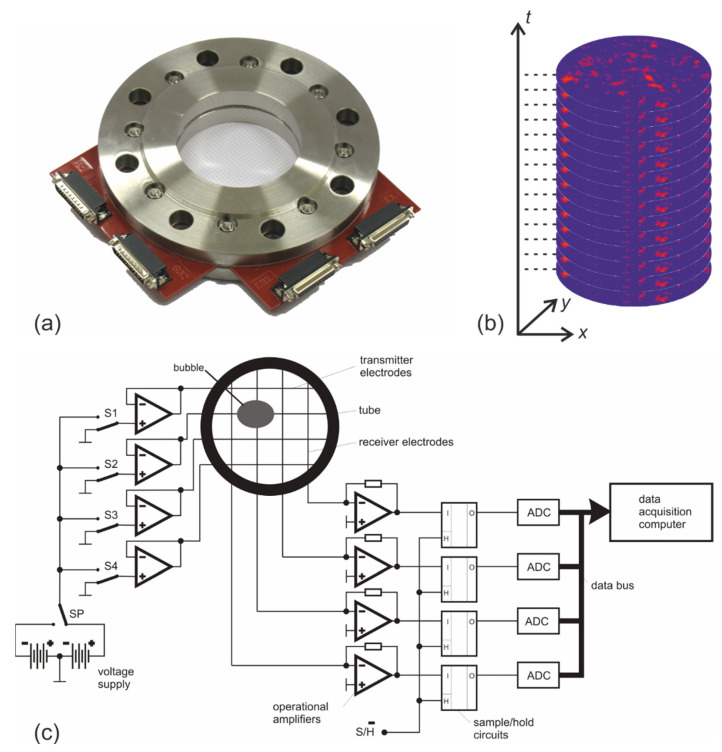
Wire-mesh sensor: (**a**) photography, (**b**) example of a 2D image sequence from a gas-liquid flow (gas phase in red color) obtained with a 2 × 64 wire-mesh sensor, (**c**) electronics scheme.

**Figure 4 sensors-22-02309-f004:**
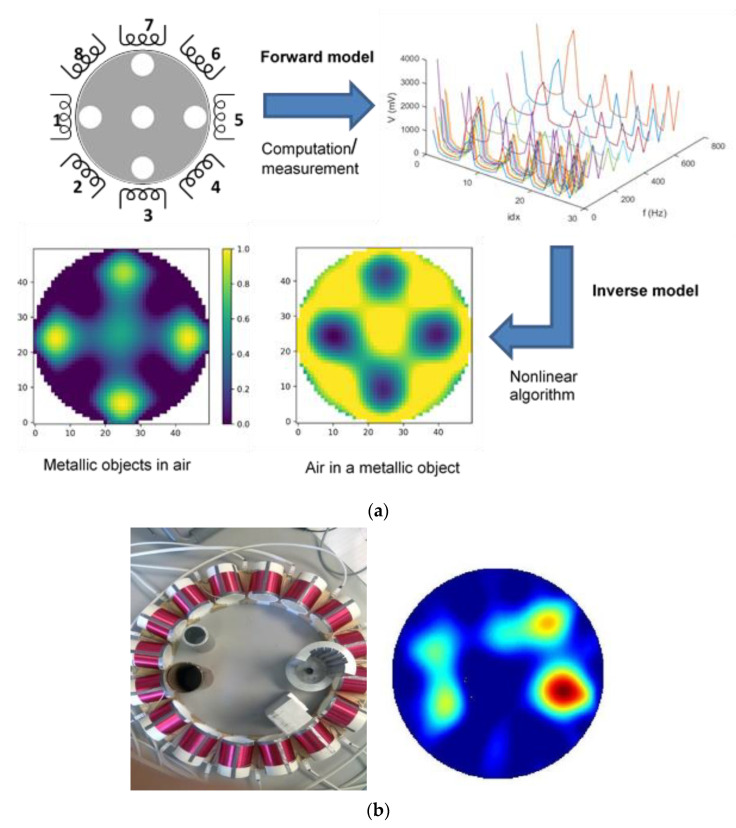
(**a**) MIT imaging process including sensor array (coils 1–8), data collection, and forward modelling, leading to measured induced voltages, 28 measured data (“Idx” denotes the measurement number index), and in various frequencies f, through inverse algorithms, PEP images can be produced. The plot shows the measured MIT data in several frequencies. (**b**) Sensor with multiple inclusions and reconstructed image, the color map represents the electrical conductivity changes.

**Figure 5 sensors-22-02309-f005:**
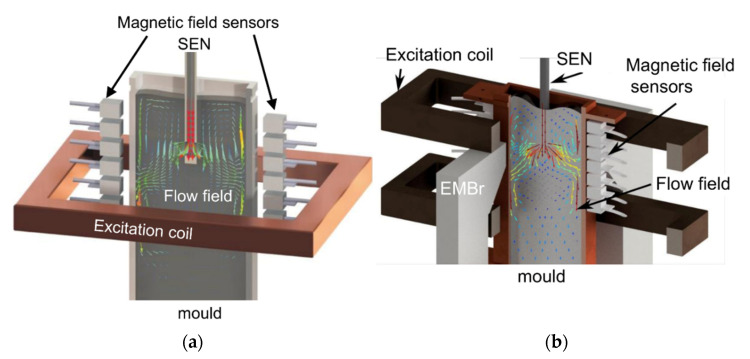
Schematic sketch of a CIFT sensor arrangement for a model of a continuous casting mold with a cross-section of 140 mm × 35 mm: (**a**) without an EMBr; (**b**) with an EMBr.

**Figure 6 sensors-22-02309-f006:**
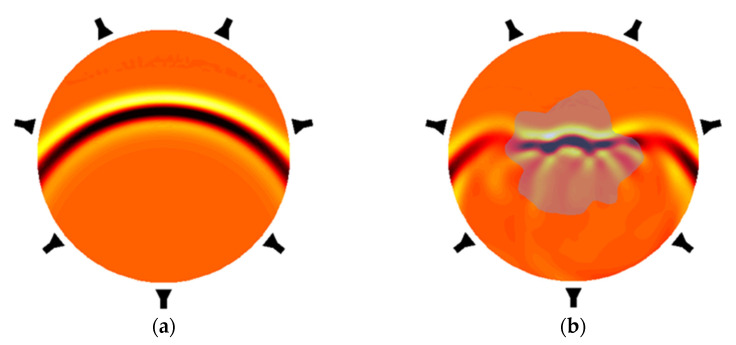
A schematic picture of the microwave propagation inside two different domains with seven antennas around the domain of interest. (**a**) Microwave field without any objects in the domain. (**b**) Microwave field for a scattering object at the center of the domain.

**Figure 7 sensors-22-02309-f007:**
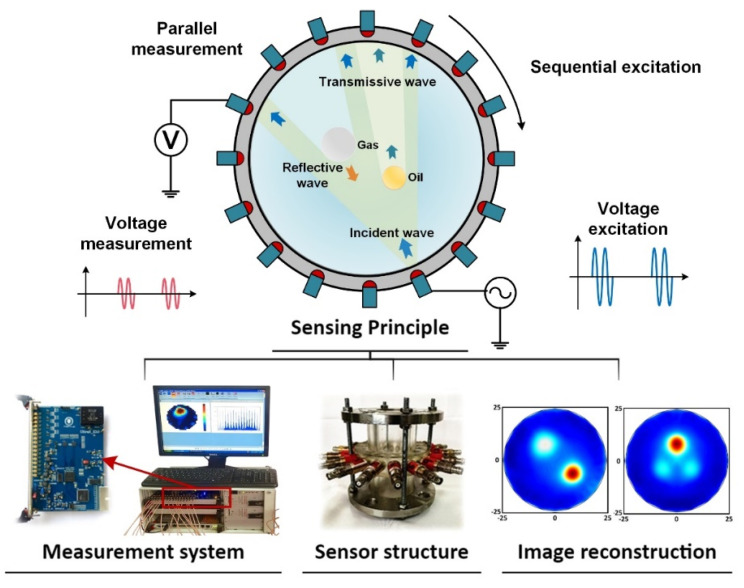
Sensing principle of UPT and image reconstruction of inclusions with different acoustic impedance.

**Figure 9 sensors-22-02309-f009:**
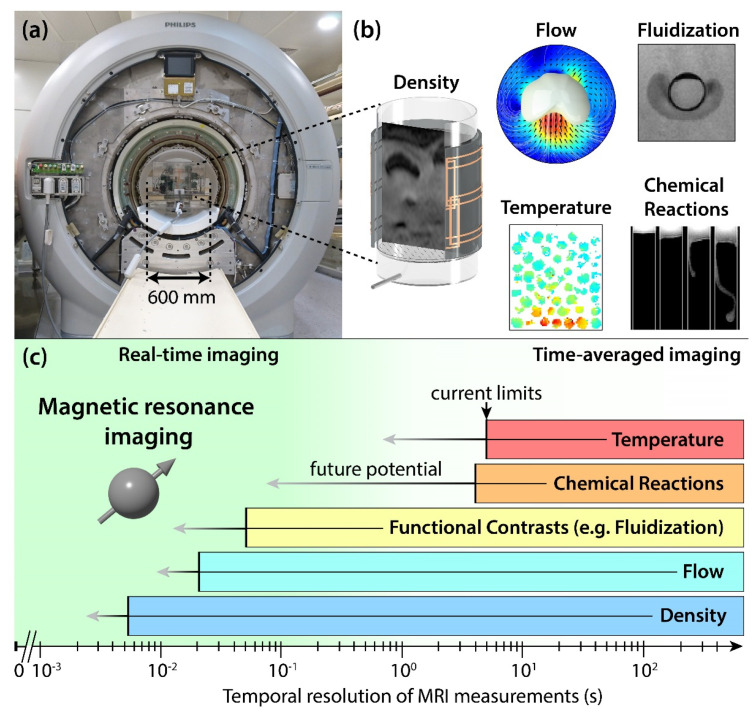
(**a**) Full body clinical 3T MRI scanner located at ETH Zurich. (**b**) MRI measurements of different sample properties. From left to right and top to bottom: Instantaneous snapshot of gas bubbles in a gas-solid bubbling fluidized bed [110], the flow of particles around a single gas bubble rising in a fluidized bed [110], cushion of locally fluidized particles observed below a single sphere sinking into an aerated particle packing [115] (reproduced with permission from Tsuji et al., *Phys. Rev. Fluids*
*6*, 064305, published by American Physical Society, 2006), temperature distribution of a particle packing heated from below by a stream of hot air [116], and an autocatalytic chemical reaction in which the reactants show a dark MRI contrast and the products a bright contrast [117] (reproduced with permission from Evans et al., *JACS*
*128*, 7309–7314, published by American Chemical Society, 2021). (**c**) Current limits of temporal resolution for different MRI measurements in process engineering systems and their expected gains using established image acceleration techniques (horizontal arrows).

**Table 1 sensors-22-02309-t001:** Summary of the properties of process tomography modalities. Note: values for temporal and spatial resolution are for “the best” existing instruments. The “type of inverse problem” indicates the usual associated reconstruction and data processing effort. Of course, direct project data processing, e.g., via machine learning approaches, may be an option to reduce computational effort as indicated in many places in this paper. However, then we would rather no longer refer to it as “tomography”.

Tomography Technique	Typical Temporal Resolution (≤)	Typical Spatial Resolution	Type of Inverse Problem	Costs	Intrusiveness and Robustness	Applications
ERT/ECT	50 fps	~5–10 mm	regularization method (e.g., Tikhonov)	low	non-intrusive	ECT: gas-solid flow, ERT: gas-liquid flow, moisture estimation dual modality: three-phase flow
Wire-mesh sensor	10,000 fps	~3 mm	none	medium	intrusive (wire electrodes in the process)	gas-liquid and liquid-liquid flows
Spectro-tomography	100 fps	~2 mm	conventional raw data available for any preferred reconstruction algorithm for each spectral point of interest	medium	usable for various modes; typically non-intrusive and robust	multi-component processes requiring information of spatial distribution with component identification
Magnetic induction tomography	10 fps	~2 mm	total variation regularization	low	non-intrusive	metallurgy and water cut metering in the oil and gas industry
CIFT	1 fps	~5–10 mm	linear inverse problem with Tikhonov regularization	medium	non-intrusive	liquid metal applications, no limit in temperature
Microwave tomography	30 fps	~5–10 mm	non-linear, ill-posed	medium	contactless, non-intrusive, and robust	medical and industrial process imaging
Ultrasound tomography	600 fps	~2 mm	total variation regularization/back projection	medium	non-intrusive	gas-liquid, liquid-liquid, liquid-solid flows
X-ray tomography	8000 fps	~1 mm	linear filtered back projection	expensive	non-intrusive	all types of flows and processes
Magnetic resonance imaging	140 fps	~0.5–3 mm	traditional MRI: discrete Fourier transform; parallel MRI: ill-conditioned inverse problem	expensive	non-intrusive	all types of flows and processes containing NMR active nuclei (^1^H, ^13^C, ^19^F)

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
