# Peer review of "A Review on Fast Tomographic Imaging Techniques and Their Potential Application in Industrial Process Control"

_sensors, 2022, doi:10.3390/s22062309_

Round 1

Reviewer 1 Report

Dear authors,

Congratulations for this very well redacted review on Industrial Process Tomography techniques. I am confident many readers will find it very instructive. The manuscript is already in an acceptable state for publishing, yet, I have made a number of minor comments (in pdf attached) you might want to consider.

Reviewer 2 Report

The article “A review on fast tomographic imaging techniques and their potential application in industrial process control” presents a review of several non-optical tomography sensors and their applications in industry.

In my opinion, the paper is valuable and well prepared. It would be nice to have a look at some of the small additions I list below.

Major remarks:

  1. There is no general diagram showing the position of the discussed methods in the areas of industrial applications (current applications, invasiveness, detection resolution, etc.).
  2. Work would benefit from showing a few examples of applications of multi-spectral tomography and CIFT. Compared to the rest of the discussed methods, the examples with the literature cited are practically absent here.

Minor:

  1. In Introduction, line 56-57, some details about the first usage of so called CT would be valuable (i.a. what processes was it accompanying). Line 67-68 - please indicate the most important literature in this field.
  2. DC/AC excitation scheme – should be written with an upper case letters. (lines, 165,171).
  3. The right plot on Figure 3 and the left diagram (a) on Figure 8 are of low quality. Please, improve it.
  4. Line 376: IPT shortcut should be clarify.
  5. Line 655: please, correct a typo in the autor Name.

Author Response

see attachement

Reviewer 3 Report

This review shows an interesting classification of research works conducted in the area of fast tomographic imaging techniques in process industry prepared by authors from different universities. The authors provide very good, systematic overview of different tomographic approaches and their applications within this area. A few comments are given as follows.

(1) Some typical review papers related with tomographic and multi-phase flows published in recent are missing in the references. . Some examples include:

* Ultrasonic Doppler Technique for Application to Multiphase Flows: A Review, Int. J. Multiphase Flow, 144(2021), 103811

* Application of electrical capacitance tomography in pharmaceutical fluidised beds – A review, Chemical Engineering Science, 2021, 231, 116236

* Application of electrical capacitance tomography in circulating fluidised beds – A review, Applied Thermal Engineering, 2020, 176, 115311

(2) One important lacuna in this review is the missing description of the different tomographic imaging challenging for the application in industrial process. For example, the sensor scale effect on the measurement accuracy which is typical interested in the real process measurement and control.

(3) For the microwave tomogrpahy, the imaging speed has been up to 30 frames/second and this should be updated in this review as reported in the literature: Chemical Engineering Science, 180(2018), 20-32.

(4) The authors gave details description of different type of tomographic imaging techniques and their application in industrial process. It is better to provide some typical measurement results and providemuch in the way of critical insightinto the results. 

Nevertheless, the review could be accepted anyway, since it will be a very interesting paper that could serve as a starting point for those looking for the multi-phase flow measurement.

Author Response

see attachement
